# Serious Games and Their Effect Improving Attention in Students with Learning Disabilities

**DOI:** 10.3390/ijerph16142480

**Published:** 2019-07-11

**Authors:** Patricia García-Redondo, Trinidad García, Débora Areces, José Carlos Núñez, Celestino Rodríguez

**Affiliations:** 1Department of Psychology, Faculty of Psychology, University of Oviedo, 33003 Oviedo, Spain; 2Department of Education, University School of Education Padre Ossó, 33008 Oviedo, Spain

**Keywords:** serious games, multiple intelligences, attention, intervention, learning disabilities

## Abstract

Previous studies have shown the positive effects of educational video games (serious games) in improving motivation, attention and other cognitive components in students with learning disabilities. This study analyzes the effects on attention of a serious game based on multiple intelligences in a sample of 44 students (age range = 6–16 years; experimental group = 24; control group = 20) with attention deficit hyperactivity disorder (ADHD) and specific learning disorder (SLD). Performance and observation measures of attention were used. The intervention consisted of 28 sessions (10 min each), in which the participants trained with 10 games based on multiple intelligences. A significant improvement in attention performance measures (visual attention) was found after the intervention, with the experimental and the control groups significantly differing in the posttest. These results invite consideration of the applicability of boosting different intelligences, talents or unique abilities through educational videogames as an important bridge to improving areas of deficit-in this case attention-in students with learning disabilities.

## 1. Introduction

Authors such as Boot, Kramer, Simons, Fabiani and Gratton [1] define the 21st century as the era of digital game-based learning (GBL). GBL is defined as “an environment where game content and game play enhance knowledge and skill acquisition, and where game activities involve problem solving spaces and challenges that provide players/learners with a sense of achievement” [2] (p. 51).

The continuous advance in new technologies poses great challenges and opportunities for today’s society. New technologies allow immediate, easy, up-to-date access to information and entertainment. The increasing generation of digital resources has led to the emergence of new ways of thinking, learning, and interacting with each other, the social, and physical environment. On the other hand, the customization of resources and services linked to the digital era is making personal characteristics—such as an individual’s abilities or talents—increasingly important in our society. One of the questions that arises is: Do new technologies have the potential to improve cognitive processes through increasingly adapted resources?

Nowadays, digital tools such as smartphones and tablets have become almost ubiquitous. They are frequently used to obtain information and for entertainment, often through videogames. Video games are one of the main entertainment options for children, young people, and adults, and have become a cultural mechanism of great social importance. According to data from the Spanish Association of Video Games [3], in 2017, more than 16 million people in Spain could be considered gamers (regular consumers of video games). The same report indicated that the video game industry reported record figures with growth of 16% in the previous year, generating 1359-million-euro profit, more than music and movies put together.

Serious games (SGs) are a category of video games that are used for educational purposes in different environments [4,5]. Though SGs share most of their technology with traditional video games, their aims and uses are outcome-driven in comparison. It is fundamental in this sense to define the objectives, content, skills, and behaviors to develop while not forgetting aesthetic, narrative, and technical resources to encourage engagement and playability, which are essential elements in a video game [6]. Authors such as Starks support the pedagogical use of video games, pointing out that they allow the introduction of evaluative and educational objectives without sacrificing entertainment, using a motivating and meaningful methodology. It is the instructional design which distinguishes a commercial videogame from a videogame with an educational focus or a pedagogical tool (a serious game). Though they share technology, they have completely different objectives and uses. In serious games, the objectives, content, assessment procedure, skills and competences to develop are well defined without forgetting the aesthetic, narrative and technical resources of videogames that encourage engagement and playability [7].

Video games are increasingly used in the field of special education to support well-being, social skills, independent living, and inclusion in varied samples of students with special needs such as autism spectrum disorders, learning disabilities, and giftedness [8,9,10,11]. According to Sánchez-Peris [12] and Sedeño [13], the use of these types of games is also an excellent way to improve attention, effort, and motivation; to develop competencies and skills such as mental agility; and to promote understanding, reflection, and strategic reasoning.

In the case of the attentional component, the effects of SGs have been examined in students with attention deficit hyperactivity disorder (ADHD) [14,15,16]. Those authors found improvements in time management and planning/organization, and a reduction in hyperactivity symptoms in a group of students with ADHD who received an SG intervention. In addition, Schubert et al. [17] found advantages in visual attention in expert video gamers compared to non-experts, especially in the domains of perception threshold and visual processing speed. These effects were not moderated by personal characteristics such as personality, intelligence, or health status.

Recent studies have found that these tools can also have positive effects on aspects such as reading skills [18,19]; vocabulary, language learning, and listening [20,21,22]; spelling [23,24]; mathematics [25]; and even affective-motivational components [26].

An important aspect to explain the positive effects reported in these studies is the fact that SG activities are motivationally challenging while simultaneously offering the students a fun learning experience. The use of attractive narratives and technical resources that are present in videogames can increase the motivation of the student to learn, increasing levels of engagement, which ensures involvement in a game. Levels of engagement are linked to positive emotions produced by effort and overcoming obstacles, which are essential aspects in turning a videogame into an educational tool (i.e., serious games) [27].

### The ToI Method: Serious Games Based on Multiple Intelligences

Recently, researchers, educationalists and engineers from Cuicui Studios, created the Tree of Intelligences (ToI) method [26,28] based on Gardner’s Theory of Multiple Intelligences (MIT) [29,30], conceived to identify and intervene in Multiple Intelligences (MI) and associated components. ToI combines the theoretical foundations of Gardner’s theory and the basic assumptions of video game design.

MIT was initially posed by Gardner [29,30], who defines intelligence as a potential which is present in the individual and can be changed through experience. The author originally identified seven intelligences: Musical, bodily–kinesthetic, logical–mathematical, linguistic, spatial, interpersonal, and intrapersonal; although he later added naturalistic intelligence to his theory.

This theory has been the subject of great interest in the educational community, as it breaks away from the unified traditional educational model. In contrast, MIT advocates a new teaching perspective centered on the individual which considers each student to be unique combinations of the different intelligences. As such, different methods, content and assessment procedures must be implemented in order to help students develop their potential [30,31,32].

Within this context, it is essential to discover the intellectual strengths and capabilities of the individual so that they can be developed from as young as possible. Once these areas are identified, they can be used as excellent foundations on which new knowledge is built, boosting the development of those areas where the student may have difficulties [30]. This is the idea behind the emergence of some interventions based on MI [9,10,33]. Specifically, Moral-Pérez et al. [33] reported some positive results in their recent study conducted in a sample of primary school students. The authors used a videogame involving playful activities based on the eight intelligences proposed by Gardner. After the intervention, an improvement in all the intelligences was found, which was statistically significant in logical–mathematical, visual–spatial and bodily–kinesthetic intelligences, evaluated through the use of questionnaires. However, the benefits of this approach seem not to be exclusively restricted to an improvement in MI components. Kuo et al. [10] analyzed the effect of a three-year school intervention based on MI with a sample of pre-school students with different profiles (gifted students, “doubly exceptional children”—gifted students with a disability such as autism or sensory disorders—and other personal conditions). At the end of the intervention, participating students demonstrated great levels of imagination, an improvement in problem solving skills, and were able to seek many approaches to solving problems. Students in the group of doubly exceptional children, especially those with autism, showed significant gains in social skills, and their group adaptability had improved. These results indicated the positive effect on students of working on MI, regardless of their needs, the nature of their talents or their cultural or socio-economic status. The authors stated that “If a teacher is having difficulty teaching a student in the more traditional linguistic or logical ways of instruction, the theory of multiple intelligences suggests several other ways in which the material might be presented to facilitate effective learning” [10].

The ToI method was created, based on previous evidence, with the aim of exploring the multiple possibilities that both serious games and the multiple intelligences approach may offer for improving cognitive components. The result of this synergy is an algorithm that allows the individual´s performance to be measured in real time, offering information about their intelligence profile and clues to strengthen their strong areas while improving their weak areas at the same time [28].

The ToI method is currently in two digital tools, “Boogies Academy” and “Cuibrain,” developed for smartphones and tablets. They are aimed at two different target audiences: Children from the age of six and adolescents, respectively. The games are therefore intended to cover two periods of special interest—late childhood and transition to adulthood. The latter is a vulnerable period of change in which adverse life events and negative outcomes can significantly affect future development [34]. This is a stage of particular interest from the viewpoint of the progression of different attentional and behavioral conditions, which, if not properly addressed in time, can have detrimental effects on subsequent development [35,36].

Each tool is made up of different games which pose a challenge (solving a problem) to the player. Depending on the skills or abilities required to solve the problem one (primary) intelligence or various (secondary) intelligences will be activated. Figure 1 shows selected screenshots of Cuibrain and Boogies Academy for smartphone and tablet.

These tools have been used in previous studies, demonstrating good psychometric properties. Garmen et al. [28] analyzed the performance of a normative sample of 372 students from first to third grade in “Boogies Academy.” Results showed a normal distribution in the variables of correct responses, playing time, and accuracy in the different games. In addition, Garmen et al. [26] analyzed the effect of both “Boogies Academy” and “Cuibrain” on levels of anxiety and self-concept in a sample of children and adolescents with learning difficulties. The results showed that participants had reduced anxiety levels while increasing self-concept after an intervention with the videogames. The authors concluded that because of its design and function, the ToI software may be an appropriate tool for the evaluation of and intervention in multiple intelligences, as well as for enhancing personal variables such as affective-motivational components.

The present study aims to analyze the effect of playing “Boogies Academy” and “Cuibrain” on attentional variables in a sample with learning difficulties, in particular ADHD and specific learning disorders (SLD). For this purpose, a quasi-experimental study with two groups (experimental and control groups) was carried out using performance-based and observation measures (questionnaires) to analyze and compare the attentional profiles of the groups. After the intervention, we expected to find significant improvement in attentional variables (assessed by means of both observation and performance-based measures) in the experimental group. Once pretest levels in attentional variables were controlled for, we also expected statistically significant differences between the control and the experimental groups after the intervention in favor of the experimental group.

## 2. Materials and Methods

### 2.1. Participants

Forty-four students with different learning difficulties (male = 27; 61.4%) took part in the study. Ages ranged from 6 to 16 years old (*M* = 11.56; *SD* = 2.67). They were recruited from an educational psychology service in Northern Spain. The mean Intelligence Quotient (IQ) was 109.12 (*SD* = 14.105). The sample was made up of students with attention deficit hyperactivity disorder (ADHD) and specific learning disorder (SLD) (Diagnostic and Statistical Manual of Mental Disorders: DSM-5; APA, 2013).

The final sample was then separated into an experimental group (N = 24), who received the MI intervention, and a control group (N = 20). The assignation to the groups was random. The characteristics of the groups were as follows:

Experimental group: 24 students (male = 14; 58.3%). Mean age 11.83 (*SD* = 2.71) and mean IQ 110.13 (SD = 14.35).

Control group: 20 students (male = 13; 65%). Mean age 11.83 (*SD* = 2.71) and mean IQ 108.16 (SD = 14.85).

There were no statistically significant differences between groups in terms of age (*p* = 0.470) or IQ (*p* = 0.665). A chi-squared (χ^2^) test indicated that gender distribution was equivalent in both the experimental (*p* = 0.414) and the control group (*p* = 0.180).

#### Inclusion Criteria

According to the DSM-5, participants were diagnosed with SLD if at least one interpretable Wechsler Intelligence Scale for Children–IV (WISC-IV) index from the verbal comprehension index or perceptual reasoning index was ≥85, and performances on reading, writing, and/or arithmetic skills were under the clinical cutoff scores indicated by cited guidelines (≤2 SD below mean performances of age-matched participants or ≤5th–10th percentile). SLD participants met A, B, C, and D criteria from DSM-5, and the level of functional impact was moderate in all cases.

The ADHD diagnosis group was confirmed by a trained researcher using the Diagnostic Interview Schedule for Children–Parent Version (DISC-P) and confirmed by the Evaluation of the Deficit of Attention and Hyperactivity scale (EDAH). Patients with any subtype of ADHD (hyperactive–impulsive, inattentive, combined hyperactive–inattentive) were eligible.

In order to select a homogeneous group of children with SLD and ADHD without significant comorbidities or potentially confounding factors (frequently co-occurring), exclusion criteria were set as the presence of other significant medical and psychological problems and comorbid disorders (e.g., developmental coordination disorder or specific language impairment) or disruptive behavior. Students who had significant cognitive, sensory, physical or emotional impairment and/or those who exhibited an IQ under 85 or over 130 were also excluded from the sample.

### 2.2. Measures

Three different measures were taken in the study. Firstly, the Wechsler Intelligence Scale for Children–IV (WISC-IV) [37] was administered in order to determine the participants´ intellectual ability. It is one of the most commonly-used scales and provides detailed information about the student’s cognitive profile. It was used to exclude those students with an IQ under 85 or over 130. Once IQ was established, attentional variables were recorded using two types of measures: Performance measures (D2 Attention Test) and observation measures (EDAH scale completed by families).

#### 2.2.1. Performance Measures

The D2 Attention Test [38] was used for the assessment of attentional variables based on the participant´s performance. This is a screening test of selective attention and concentration. It lasts about 8–10 min. The task consists of the identification of relevant stimuli (the letter d “with two stripes”). The test is composed of 14 rows (with 47 letters each row) in which the participant has to identify the relevant stimuli. The participant spends 20 s on each row. Different attention indicators are recorded: Total responses (TR), total correct responses (CR), errors of omission (O), and commissions (C). On the basis of these variables, two general measures are obtained: A measure of general performance—or quality of attention—(TOT = total responses minus the sum of errors of omission and commission) and a specific measure of sustained attention or concentration (CON = total correct responses minus commissions). Raw scores and percent rank scores are provided in the task. In the current study, the following indicators were used as dependent variables, based on raw scores: D2 quality of attention (D2-TOT), D2 concentration (D2-CON), and D2 correct responses (D2-CR).

#### 2.2.2. Observation Measures

The EDAH scale [39] was completed by families in order to assess behaviors related to attention deficit and hyperactivity/impulsivity and confirmed the selection criteria for the ADHD group. This scale evaluates attentional symptomatology described in DSM-5 [40] through the administration of a 20-item observation scale. The scale is frequently used as a screen for ADHD and helps distinguish the different presentations of the disorder: Predominantly impulsive–hyperactive, predominantly Inattentive, and the combined presentation. The raw scores in the sub-scales of attention deficit (EDAH-AD), hyperactivity/impulsivity (EDAH-H/I), and ADHD (EDAH-ADHD) were used as dependent variables in this study.

### 2.3. Procedure

This study was conducted in accordance with the Code of Ethics of the World Medical Association (Declaration of Helsinki), which establishes the ethical principles for research involving humans [41]. Participants and their parents gave written informed consent and volunteered for the study. Once written consent was obtained, the corresponding tests were conducted to verify the diagnosis and to participate in this research. The study was approved by the Ethics Committee of the Principality of Asturias (reference: CPMP/ICH/70/19, code: vRTI_Learning).

Participants were selected by means of convenience procedures, and their specification in terms of previous diagnosis was established according to the information provided by pediatric and/or school counseling services. Diagnoses were established according to a comprehensive evaluation of students’ cognitive, affective, and attentional components, as well as basic reading, writing, and mathematics skills; this was done using standardized tests. The students were recruited from the same educational psychology service in Northern Spain. Before starting the intervention, both the experimental and control groups completed the WISC-IV and the D2 Attention Test, while the EDAH scale was administered to families (pretest). D2 and EDAH were also administered at the end of the intervention (posttest).

The intervention consisted of a 28-session program (2 ten-minute sessions per week). The tools used were “Boogies Academy” and “Cuibrain,” depending on the age of the participant. For students aged between 6 and 10, “Boogies Academy” was used, while participants aged between 11 and 16 played “Cuibrain.”

The design and development of the games used a methodology called the Tree of Intelligences (ToI) method. This methodology, applied to attention deficit (Figure 2), is based on Gardner’s conception of the human mind, which is that the different intelligences work in a coordinated manner [30,42] and can be triggered by information presented both internally and externally. These intelligences, predispositions, or talents can help improve students’ weak areas. In this case, the sample was made up of students who have attention problems. Because attention is a basic process and the games used in the study address the attentional component in their mechanics, one would expect students to improve their attention using their strong abilities (different intelligences) as an access route.

The technologies developed using this method have implications for both assessment and intervention, and this is mediated by the design itself and technical characteristics of the game, which make the videogame a serious game.

The result is an algorithm which allows the real-time measurement of a player’s achievement, providing information about their profile of intelligences, as well as advice for improving strong areas and compensating for weaker areas. Both games have a total of 10 sub-games, covering at least one key ability from each of the 8 intelligences recognized by Gardner´s theory (musical, bodily-kinaesthetic, logical–mathematical, linguistic, spatial, interpersonal and intrapersonal, and naturalistic intelligence).

The main instructional strategy behind this methodology is problem solving, since the player exercises and rehearses deploying their prior skills and knowledge. In response to the concept of intelligence presented by MIT, the subjects must give the correct solution to the challenge presented by each game.

The intervention was carried out by the same member of the research team each time. Participant attendance to each session was registered. The intervention took place at the same educational psychology center participants were recruited from—a psychological center specializing in learning difficulties. During each session, participants played two games (5 min each). The games were randomly assigned to the participants each day, ensuring that each participant had played the same number of sessions per game at the end of the intervention program. Participants played the games individually (each student had a tablet) in a shared space for small groups of 4–5 children—a separate room with appropriate environmental conditions. The intervention was carried out after school hours. All the games included a built-in tutorial on how to play. No prior knowledge on multiple intelligences or the different skills put into place in each game was required.

The students were assigned to the experimental and control groups randomly. The control group did not receive any parallel or alternative training. However, for ethical reasons, they were given the same intervention once the posttest assessment was completed.

### 2.4. Data Analysis

In order to meet the objectives of the study, data were analyzed in three steps. First, to verify that the data was appropriate for parametric analyses, descriptive statistics for the dependent variables (performance and observation measures; pre and posttest) were analyzed, with special attention to skewness and kurtosis values. As the variables showed a normal distribution, parametric analyses were performed. Second, to determine the effect of the intervention, group differences in posttest were analyzed, taking possible differences in pretests and the effect of age into consideration. Separate Analyses of the Covariance (ANCOVAs) were performed, controlling for the effect of the pretest and age, in each variable, as covariates. Lastly, an estimation of the effect size (Eta squared; η^2^) was included. Based on Cohen´s [43] correspondence criterion, η^2^ = 0.01 was interpreted as small, η^2^ = 0.06 as medium, and η^2^ = 0.14 as large.

SPSS 24 was used for data analysis, establishing *p* < 0.05 as the criterion for statistical significance.

## 3. Results

### Change in Attentional Variables after the Intervention

Table 1 shows descriptive statistics for the different groups in each of the variables examined, as well as between-group differences in posttest.

The means of the performance measures of attention (D2) indicated a general improvement in each group in the posttest, with an increase in the total effectiveness of the test (TOT), concentration (CON), and accuracy, established in terms of correct responses (CR). This increase was greater in the experimental group (Figure 3). A statistical analysis of between-group differences showed that, although the groups did not differ significantly in the pretest, there were statistically significant differences in the posttest in all variables, favoring the experimental group. Effect sizes were large, with the highest effect size in CR. The covariate age did not have a statistically significant effect in any of the comparisons made.

In the case of observation measures (EDAH administered to families), there were no statistically significant differences between the groups either at pretest or posttest. Looking at each group separately, the means indicate that there was no significant reduction in the symptoms over time.

## 4. Discussion

Today, videogames have become indispensable entertainment for children and adults. At the same time, the Theory of Multiple Intelligences has progressively gained popularity, although critical positions in terms of its existence and conceptualization are also present. However, there is little research focusing on the relationship between videogames, multiple intelligences, and learning processes to date.

With respect to attentional variables assessed by means of performance measures (D2), the results indicated a general improvement in both the experimental and control groups after the intervention, with a general increase in concentration and accuracy. Looking at between-group differences, the groups were statistically significantly different in their levels of attention at the end of the intervention, with participants in the experimental group exhibiting significantly higher levels of attention in comparison to the controls. The largest effect was found in the variable correct responses, which represents performance accuracy. These results are in line with previous research stating that playing videogames can increase concentration and other cognitive variables, such as processing speed or visual discrimination [12,13,14,15,16].

Bul et al. [15], for instance, reported improvements in time management and planning/organizing, as well as a reduction in hyperactivity symptoms in a group of students with ADHD who played an SG intervention. The same authors found positive effects of SGs on attention in previous research [14] when they analyzed the effect of a 10-week SG intervention (called “Plan-It Commander”) in a sample of ADHD students aged 8–12. They also found significantly greater differences in the experimental group in time management skills and the social skill of responsibility (as reported by parents) compared to the control group. In the current study, no differences in observation measures (EDAH scale administered to families) were found. Our participants in the experimental and control groups did not differ in the variables of attention deficit, hyperactivity/impulsivity, or the combination of both—either before or after the intervention.

The lack of statistically significant differences in observation measures in our study, even when they were found in performance measures, could be explained by the different nature of the two types of attentional measures that were used. Previous research has highlighted the existence of a low correspondence between the scores of children and adolescents in traditional performance tests and the difficulties observed in various areas of daily life functioning, such as school or home, reported by different informants—particularly parents and/or teachers [44,45,46,47,48]. These studies have noted the presence of low-to-moderate associations between the information obtained by different methods or informants.

Within this context, changes in attentional performance were expected, given the mechanisms involved in traditional performance measures. In this case, the D2 test consists of discriminating visual stimuli within a context, having a reduced amount of time (20 s per row of 47 stimuli—or possible target). This task requires visual speed, concentration, and discrimination skills, which are commonly trained, explicitly or implicitly, by playing video games. This finding is in line with the study by Schubert et al. [17], who found differences in visual attention and visual processing speed between expert video gamers and non-experts.

On the whole, preliminary results from this study suggest the potential usefulness of the two videogames tested improving attention variables (in the case of the D2 test) in the current sample of children a adolescents. An increase in attentional variables was found in both groups, which was expected considering that the sample involved in the study had a prior diagnosis, and as a result, were receiving support in educational and/or clinical contexts. However, once the sample was controlled for the possible effect of pretest differences, the gain was greater in the case of the experimental group than in the control group. This is in line with the MI approach from Gardner [29,30], as well as the design and foundation behind the ToI method, according to which an individual’s profile of intelligences can provide important clues to help strengthen their strong areas and compensate for their weak points—in this study, that weak point was attention.

## 5. Conclusions

The main conclusion arising from these findings is the need to broaden the study of educational videogames and their possible benefits to different cognitive variables and diverse populations, especially those with difficulties in the automation or control of cognitive processes, such as attention. The great popularity of these tools makes them exceptional alternatives for applications with intervention and therapeutic objectives, as long as they are well defined in terms of objectives to achieve, content to work with, assessment procedures, and skills and competences to develop [7,33]. Their benefits lie in the potential to create more realistic and interactive environments that promote cognitive processes through increasingly adapted resources that can be progressively integrated with the variety of intervention approaches available today, mainly pharmacological and behavioral interventions. For instance, the use of game-based neurofeedback systems, in combination with pharmacological support, has been shown to help improve executive control in subjects with ADHD to a greater extent than pharmacological support alone [49].

Finally, there are some limitations in the study that should be acknowledged. Frist, the small sample size and the heterogeneous nature of the sample must be taken into consideration in relation to the generalization and scope of the findings. The absence of a control group of students without learning disabilities is also a limitation. Additional studies, controlling for the presence or absence of a diagnosis, are needed in order to better determine the potential of the intervention. Along similar lines, widening the sample by considering additional diagnostic groups, such as autistic spectrum disorder or students with affective-motivational disorders would be interesting, as previous research indicates that SGs can have potential effects on variables such as motivation, affect, and cognitive flexibility. In addition, the possible effect of the game must be considered. Both games, “Boogies Academy” and “Cuibrain” share the ToI methodology, and the mechanics of the games are similar. However, the aesthetics vary. “Boogies” is intended for children, and “Cuibrain” is intended for adolescents. “Cuibrain” is more difficult than “Boogies” in order to adapt to the attentional demands of the participants´ developmental levels. The possible effect of both variables (aesthetics and difficulty) must be analyzed in future studies. When it comes to effects on attention found in the study, the number of attentional measures could be increased in future studies, especially in the case of observation measures. While one of the findings from the study was that there is no correspondence between the results from performance and observation measures, including reports from other informants such as teachers would have helped to better determine any such correspondence and possible sources of the lack of agreement. Lastly, as follow-up measures were not taken either in the experimental or the control group, the long-term effects of the intervention could not be analyzed. Further research is necessary in order to better determine the potential of new technologies—especially SGs—for learning and rehabilitation.

## Figures and Tables

**Figure 1 ijerph-16-02480-f001:**
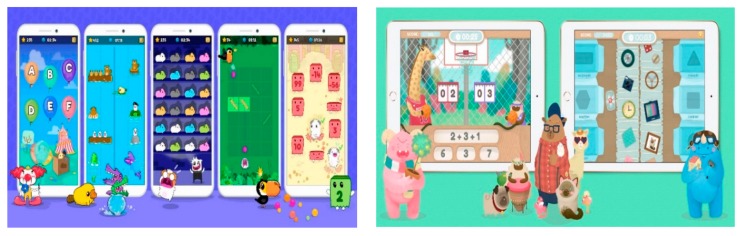
Screenshots of “Cuibrain” (left) and “Boogies Academy” (right) for smartphone and tablet.

**Figure 2 ijerph-16-02480-f002:**
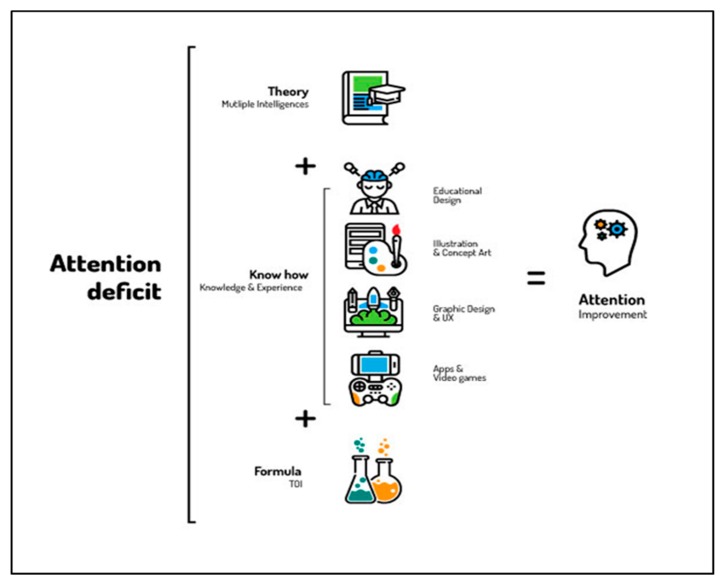
Graphic representation of the Tree of Intelligences (ToI) method applied to attention intervention.

**Figure 3 ijerph-16-02480-f003:**
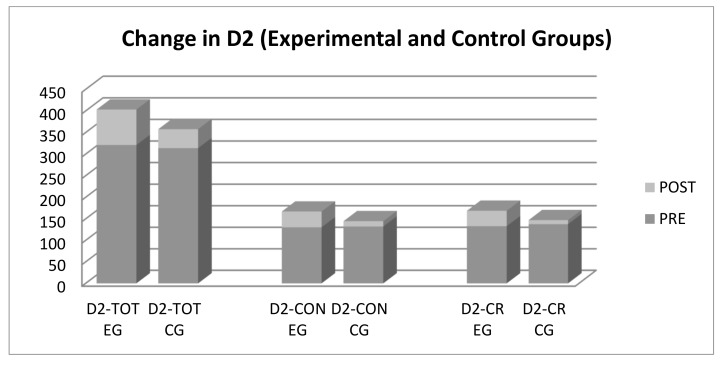
Change in attentional variables (D2 test) over time. PRE = Pretest; POST = Posttest, EG = Experimental group; CG = Control Group.

**Table 1 ijerph-16-02480-t001:** Descriptive statistics (pre and posttest) and between-group differences in D2 and EDAH.

		Experimental Group (*n* = 24)		Control Group (*n* = 20)		Differences	
		M	SD	M	SD	*p*	η^2^
Performance measures							
D2-TOT	PRE	321.05	75.71	313.81	76.03	0.754	0.002
	POST	403.25	102.02	357.80	91.45	0.013	0.145
D2-CON	PRE	130.40	31.77	132.33	49.75	0.877	0.001
	POST	166.97	45.64	144.56	37.17	0.009	0.157
D2-CR	PRE	132.97	31.69	137.71	49.82	0.704	0.003
	POST	168.69	45.23	147.07	36.15	0.002	0.210
Observation measures							
EDAH-AD	PRE	7.06	3.198	7.60	3.033	0.573	0.008
	POST	7.23	2.859	7.23	2.859	0.668	0.005
EDAH-H/I	PRE	6.75	3.674	7.40	3.424	0.550	0.009
	POST	5.98	3.116	6.78	3.636	0.881	0.001
EDAH-ADHD	PRE	13.21	5.703	15.00	5.794	0.309	0.025
	POST	13.21	5.099	13.61	3.306	0.999	<0.001

Note. D2-TOT = Total-quality of attention; D2-CON = Concentration; D2-CR = Correct responses; EDAH-AD = Attention deficit symptoms; EDAH-H/I = Hyperactivity/impulsivity symptoms; EDAH-ADHD = Attention deficit and hyperactivity/impulsivity symptoms; PRE = Pretest; POST = Posttest.

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
