# Peer review of "Serious Games and Their Effect Improving Attention in Students with Learning Disabilities"

_ijerph, 2019, doi:10.3390/ijerph16142480_

Round 1

Reviewer 1 Report

This is, in summary, an interesting manuscript aimed to explore the effects on attention of a serious game based on multiple intelligences in a sample of 44 students (experimental group=24; control group=20) with Attention Deficit Hyperactivity Disorder (ADHD) and Specific Learning Disorder (SLD). The authors reported that a significant improvement in attention performance measures (visual attention) was found after the intervention, with the experimental and the control groups significantly differing in the post-test.

The authors may find as follows my main comments/suggestions.

First, as throughout the Introduction section the authors correctly focused on young students as particiapnts of their study, they should even refer to adolescence as a vulnerable period of changes that identifies the transition from childhood to adulthood. Importantly, the link between adverse life events and negative outcomes in young adolescents may deserve to be, at least rapidly, mentioned. According to the main findings of a systematic review, the number of the experienced adversities or negative life events seemed to have a positive dose-response relation with youth suicidality. Thus, although i understand that the link between adolescence and negative outcomes is not the main topic of the present manuscript, i suggest to cite within the main text the paper published on Eur Child Adolesc Psychiatry in 2015 (PMID: 26303813).

In addition, as the authors reported extensively the most important aims/objectives of this paper, the main study hypotheses should be similarly described in a more detailed manner.

Moreover, the inclusion/exclusion criteria need to be better specified.

Also, the rationale underlying the choice of the specified psychometric instruments (e.g., he Wechsler Intelligence Scale for Children–IV scale, EDAH) needs to be clearly described.

Furthermore, within the first lines of the Discussion section, the authors do not need to repeat again what are the most relevant aims of this paper with regard to the main topic, as these objectives have been already discussed extensively elsewhere. Here, i suggest to immediately focus on the most relevant findings of the study and their implications for the general readership.

Finally, what is the take-home message of this manuscript? While the authors claimed the applicability of boosting different intelligences as well as unique abilities through educational videogames as bridges to improve areas of deficit -in this case attention- in students with learning disabilities, they failed, in my opinion, to provide some cocnlusive remarks about their topic. Specifically, how educational videogames may be used in the clincial practice in order to promote cognitive processes through increasingly adapted resources? How new technologies may be integrated to existing pharmacological and behavioral interventions? Here, more details/information are needed to this specific regard.

Author Response

The authors acknowledge he comments from the reviewers and the possibility to submit an improved version of the manuscript. Below, we respond to the comments made by each reviewer.

Along with the response to the different comments and suggestion, we included the ethical approval number in procedure section (lines 226-227). English language was also edited by professionals.

Reviewer 1

This is, in summary, an interesting manuscript aimed to explore the effects on attention of a serious game based on multiple intelligences in a sample of 44 students (experimental group=24; control group=20) with Attention Deficit Hyperactivity Disorder (ADHD) and Specific Learning Disorder (SLD). The authors reported that a significant improvement in attention performance measures (visual attention) was found after the intervention, with the experimental and the control groups significantly differing in the post-test.

The authors may find as follows my main comments/suggestions.

1.First, as throughout the Introduction section the authors correctly focused on young students as particiapnts of their study, they should even refer to adolescence as a vulnerable period of changes that identifies the transition from childhood to adulthood. Importantly, the link between adverse life events and negative outcomes in young adolescents may deserve to be, at least rapidly, mentioned. According to the main findings of a systematic review, the number of the experienced adversities or negative life events seemed to have a positive dose-response relation with youth suicidality. Thus, although i understand that the link between adolescence and negative outcomes is not the main topic of the present manuscript, i suggest to cite within the main text the paper published on Eur Child Adolesc Psychiatry in 2015 (PMID: 26303813).

A reference to the recommended paper was included (Reference number 34: lines 126-128 and in reference list).

2.In addition, as the authors reported extensively the most important aims/objectives of this paper, the main study hypotheses should be similarly described in a more detailed manner.

The main study hypotheses were clarified (page 4, lines 153-157).

3.Moreover, the inclusion/exclusion criteria need to be better specified.

Inclusion/exclusion criteria were specified in a new sub-heading 2.1.1 Inclusion criteria (pages 4 and 5, lines 175-191).

4.Also, the rationale underlying the choice of the specified psychometric instruments (e.g., he Wechsler Intelligence Scale for Children–IV scale, EDAH) needs to be clearly described.

The election of the Wechsler scale is justified on page 5, lines 193-196; and EDAH on page 5, lines 213-216 .

5.Furthermore, within the first lines of the Discussion section, the authors do not need to repeat again what are the most relevant aims of this paper with regard to the main topic, as these objectives have been already discussed extensively elsewhere. Here, i suggest to immediately focus on the most relevant findings of the study and their implications for the general readership.

The objective was eliminated from the discussion part (page 8).

6.Finally, what is the take-home message of this manuscript? While the authors claimed the applicability of boosting different intelligences as well as unique abilities through educational videogames as bridges to improve areas of deficit -in this case attention- in students with learning disabilities, they failed, in my opinion, to provide some cocnlusive remarks about their topic. Specifically, how educational videogames may be used in the clincial practice in order to promote cognitive processes through increasingly adapted resources? How new technologies may be integrated to existing pharmacological and behavioral interventions? Here, more details/information are needed to this specific regard.

A more extended explanation of the practical implications of the study was included (pages 9 and 10, lines 374-379).

Reviewer 2 Report

This quasi-experimental study analyzes the effects on attention of a serious game (“Boogies Academy” and “Cuibrain”) based on multiple intelligences in a sample of 44 students (age range= 6-16 years; experimental group= 24; control group= 20) with Attention Deficit Hyperactivity Disorder (ADHD) and Specific Learning Disorder (SLD). This study found a significant improvement in performance measures of visual attention after the intervention, with the statistically significant differences in the posttest, favoring the experimental group. However, there was no significant reduction in the symptoms on the observation measures over time.

As digital game-based learning becomes more and more common and important nowadays, the present study provides knowledge to this field. I would like to suggest the authors doing some revisions:

1.          The participants had a wide range of age distribution (age range = 6-16). Although the small sample size limited the possibility to examine age effect on the effect of the serious game, the authors need to address the possible influence of age.

2.          The participants had ADHD and SLD. The diagnoses and severities of SLD should be introduced.

3.          “DSM-V” should be corrected into “DSM-5.”

4.          Line 167-168: “Gender distribution was equivalent in both the experimental (p = .414) and the control group (p = .180).” It is unclear what comparison was made.

5.          Line 170 and 248: “Different” was used here. However, the meaning is unclear.

6.          The paragraphs in Line 223 to 246 described the design and development of “Boogies Academy” and “Cuibrain.” They should be moved to Line 209 to make the introduction of intervention clearly and comprehensively.

7.          Line 293-295 could be deleted.

8.          The authors may add some discussion about the possible differences in the effects between “Boogies Academy” and “Cuibrain” designed based on multiple intelligences and videogames for entertainment.

Author Response

The authors acknowledge he comments from the reviewers and the possibility to submit an improved version of the manuscript. Below, we respond to the comments made by each reviewer.

Along with the response to the different comments and suggestion, we included the ethical approval number in procedure section (lines 226-227). English language was also edited by professionals.

Reviewer 2

This quasi-experimental study analyzes the effects on attention of a serious game (“Boogies Academy” and “Cuibrain”) based on multiple intelligences in a sample of 44 students (age range= 6-16 years; experimental group= 24; control group= 20) with Attention Deficit Hyperactivity Disorder (ADHD) and Specific Learning Disorder (SLD). This study found a significant improvement in performance measures of visual attention after the intervention, with the statistically significant differences in the posttest, favoring the experimental group. However, there was no significant reduction in the symptoms on the observation measures over time. As digital game-based learning becomes more and more common and important nowadays, the present study provides knowledge to this field. I would like to suggest the authors doing some revisions:

1.The participants had a wide range of age distribution (age range = 6-16). Although the small sample size limited the possibility to examine age effect on the effect of the serious game, the authors need to address the possible influence of age.

New ANCOVAs were performed, taking pretest scores and age as covariates. The effect of age was not statistically significant in any case and posttest results did not change (page 7 - data analysis, lines 283-285; page 8 – results, lines 305-306),

2.The participants had ADHD and SLD. The diagnoses and severities of SLD should be introduced.

Inclusion/exclusion criteria for SLD and severity were specified in a new sub-heading 2.1.1 Inclusion criteria (pages 4 and 5, lines 175-191).

3.“DSM-V” should be corrected into “DSM-5.”

This mistake was corrected (page 4, line 164).

4.Line 167-168: “Gender distribution was equivalent in both the experimental (= .414) and the control group (= .180).” It is unclear what comparison was made.

A mention to the statistic used was included (page 4, line 173).

5.Line 170 and 248: “Different” was used here. However, the meaning is unclear.

Line 170 (now line 193): “different” was referring to the three measures that were used in the study. This aspect has been clarified in the text.

Line 248 (now line 278): different” was referring to the three phases in which data were analyzed.  This aspect has been clarified in the text.

6.The paragraphs in Line 223 to 246 described the design and development of “Boogies Academy” and “Cuibrain.” They should be moved to Line 209 to make the introduction of intervention clearly and comprehensively.

This paragraph was moved to lines 223-230 (line 209 of the previous version of the manuscript).

7.Line 293-295 could be deleted.

The repetition of the aim of the study was deleted from the discussion part (page 8).

8.The authors may add some discussion about the possible differences in the effects between “Boogies Academy” and “Cuibrain” designed based on multiple intelligences and videogames for entertainment.

A mention to the possible differential effect of the games is made in the discussion taking this aspect as a possible limitation (lines 388 to 393).

Reviewer 3 Report

This paper is focused on an important subject. Hence as a non-English non-gamer reader, it was difficult to follow the aim and justification which were presented in the text. I suggest the title to be amended. From my point of view, since the focus of the paper is on the video game putting a term such serious game which implies to a special type of video game as the main category in the title of the paper might not be very helpful. But if the commercial and serious games are really two different formats of games (as it is stressed in the text) the difference and the reason for focusing on a special type of game must be forwarded to the beginning of the text instead of line 120  and the end of the review of the literature part.

Children with special needs are mentioned here with different terms such as students with special needs and exceptional children. I suggest picking one single term to cover all children with developmental disabilities or atypically developing children.

In line 158 (DSM-V; APA, 2013) DSM-V should change to DSM-5 since the Roman number in the new format of DSMs changed to ordinary numeric one.

More information is needed regarding inclusion and exclusion criteria for sample recruitment

It is also important to have some information about the diagnosis process of ADHD and SLD group. How did they receive a diagnosis? Which scale was used? Who did the diagnosis? And since these groups are heterogeneous reporting information on the level of severity of symptoms could be very helpful.

I also suggest adding some information about the children’s follow-up information on the longitudinal impacts on the experimental group or the impacts of post-study video game training on control group as it is mentioned to justify the ethical consideration for the control group (line 220 to 225 -The students were assigned to the experimental and control groups randomly. The control group did not receive any parallel or alternative training. However, for ethical reasons, they were given the same intervention once the posttest assessment was completed.) This information might act as a triangulation approach for the presented findings.

Author Response

The authors acknowledge he comments from the reviewers and the possibility to submit an improved version of the manuscript. Below, we respond to the comments made by each reviewer.

Along with the response to the different comments and suggestion, we included the ethical approval number in procedure section (lines 226-227). English language was also edited by professionals.

Reviewer 3

This paper is focused on an important subject. Hence as a non-English non-gamer reader, it was difficult to follow the aim and justification which were presented in the text. I suggest the title to be amended. From my point of view, since the focus of the paper is on the video game putting a term such serious game which implies to a special type of video game as the main category in the title of the paper might not be very helpful. But if the commercial and serious games are really two different formats of games (as it is stressed in the text) the difference and the reason for focusing on a special type of game must be forwarded to the beginning of the text instead of line 120  and the end of the review of the literature part.

The difference between serious games and commercial games is now placed at the beginning of the introduction (page 2, lines 54 to 59).

1.Children with special needs are mentioned here with different terms such as students with special needs and exceptional children. I suggest picking one single term to cover all children with developmental disabilities or atypically developing children.

The term “exceptional” is only used in the context of the study by Kuo et al. (2010) (page 3, line 109), referring to “doubly exceptional children” or those gifted students with a disability such as Autism or sensory disorders. This term is not intended to be interchangeable with the term “special needs”, which is used on some ocassions in the text.

2.In line 158 (DSM-V; APA, 2013) DSM-V should change to DSM-5 since the Roman number in the new format of DSMs changed to ordinary numeric one.

This mistake was corrected (page 4, line 164).

3.More information is needed regarding inclusion and exclusion criteria for sample recruitment.

Inclusion/exclusion criteria were specified in a new sub-heading 2.1.1 Inclusion criteria (pages 4 and 5, lines 175-191).

4.It is also important to have some information about the diagnosis process of ADHD and SLD group. How did they receive a diagnosis? Which scale was used? Who did the diagnosis? And since these groups are heterogeneous reporting information on the level of severity of symptoms could be very helpful.

A brief description of the diagnostic procedure is provided on pages 5 and 6 (lines 228 to 235). Also, Inclusion/exclusion criteria for SLD and severity were specified in a new sub-heading 2.1.1 Inclusion criteria (pages 4 and 5, lines 175-191).

5.I also suggest adding some information about the children’s follow-up information on the longitudinal impacts on the experimental group or the impacts of post-study video game training on control group as it is mentioned to justify the ethical consideration for the control group (line 220 to 225 -The students were assigned to the experimental and control groups randomly. The control group did not receive any parallel or alternative training. However, for ethical reasons, they were given the same intervention once the posttest assessment was completed.) This information might act as a triangulation approach for the presented findings.

Although the intervention was implemented in the control group, follow-up measures were not taken. This aspect was highlighted as a limitation of the present study (lines 398 to 401).

Round 2

Reviewer 1 Report

In the revised manuscript, the authors addressed successfully most of the major comments raised by Reviewers improving the main structure of the main text. I have no further additional questions.

Author Response

The authors acknowledge all the comments from the reviewer, which have significantly contributed to improve the quality of the manuscript.

Reviewer 3 Report

This is an improved version of the previously reviewed manuscript. The authors tried to address the issues raised by other reviewers as well as those which I encountered during the review.